# The Phytoremediation Potential of 14 *Salix* Clones Grown in Pb/Zn and Cu Mine Tailings

Chenggang Jiang [1,2], Yangdong Wang [2], Yitai Chen [2], Shufeng Wang [2], Changcheng Mu [1] and Xiang Shi [2,*]

1   School of Forestry, Northeast Forestry University, Harbin 150040, China; 18267026265@163.com (C.J.)
2   Key Laboratory of Tree Breeding of Zhejiang Province, Research Institute of Subtropical Forestry, Chinese Academy of Forestry, Hangzhou 311400, China
*   Correspondence: risf4017@outlook.com

**Abstract:** Willows are suitable candidates for phytoremediation projects. A pot experiment was conducted to evaluate the potential of using *Salix* unrooted cuttings for the phytoremediation of lead/zinc (Pb/Zn) and copper (Cu) mine tailings. Cuttings of 14 *Salix* clones were directly rooted into pots containing mine tailings. The 14 clones showed different levels of tolerance to tailing treatments. A total of 71.40% and 85.70% of the *S. jiangsuensis* '172' cuttings either grown in Pb/Zn or Cu tailings survived, respectively. However, the other clones had lower survival rates, and the values were no more than 40%. Usually, all clones produce less biomass in an extremely contaminated environment. Clonal variation in biomass yield was observed in this research. The surviving clones, such as *S. integra* 'WSH', *S. matsudana* '14', *S. chaenomeloides* '3', *S. chaenomeloides* '4', and *S. chaenomeloides* '5' (Pb/Zn tailing), *S. integra* 'HY', *S. integra* 'WSH', *S. matsudana* '14', *S. matsudana* '19', and *S. matsudana* '34' (Cu tailing) produced relatively more biomass in this study. In general, all the clones presented lower bioconcentration factor values for the tailings of heavy metals. In principle, all clones could easily take up and translocate Zn and Cd from the tailings to aboveground parts, especially *S. integra*. All clones exhibited a huge variation in their heavy metal accumulation capacity. As stated above, the direct utilization of cuttings for phytoremediation is a viable option. *S. jiangsuensis* '172' had a high tolerance capacity and would be a recommended candidate for future phytoremediation projects in soils containing tailings with an extremely high concentration of heavy metals. These results provide crucial information about willow growth and metal accumulation capacity in extremely adverse environments.

**Keywords:** *Salix* clones; cuttings; phytoremediation; variation; tailings; evaluation methods

## 1. Introduction

Metals are widely utilized in various aspects of life and play a crucial role in human activities and societal progress. However, during the process of mining operations and smelting, potentially hazardous trace elements are generated [1]. Consequently, surrounding soils become contaminated with heavy metals [2–4]. Moreover, the practice of tailings revegetation becomes challenging due to a limited water retention capacity, nutrient deficiency, and elevated metal concentrations [5]. The presence of high concentrations of heavy metals exerts detrimental effects on both plants and soil microorganisms, as has been well established in the scientific literature [6,7]. Furthermore, heavy metals can pose potential risks to both humans and animals through the food chain [8–10]. Therefore, there is an urgent need for ecological remediation of mine tailings [11].

Phytoremediation represents a green and low-cost technology for ecological restoration in mining areas [3,12,13]. Using phytoremediation is becoming more widely recognized as a potential technique for soil remediation [4]. The careful selection of plant species is a critical factor contributing to the success of revegetation efforts [10,14]. Fast-growing plants that produce high amounts of biomass—characterized by deep rooting systems, high

transpiration rates, strong resilience to barren conditions, and tolerance to excess metal concentrations, and which possess the ability to accumulate high levels of metals in harvestable portions—are the optimal plant species for successful phytoremediation [15–17]. Numerous studies have suggested that fast-growing species, such as poplar and willow, are commonly employed in phytoremediation [17–21], owing to their high biomass productivity and adaptability to adverse conditions [22]. Many investigations have suggested that *Salix* spp. can normally grow on moderately contaminated soils [23,24], even in mine tailings with exceptionally high concentrations of heavy metals [25,26]. Furthermore, numerous willow species/clones have demonstrated their phytoremediation potential in both field and greenhouse experiments [27–31]. Therefore, these studies have provided insights into the potential and variability of willow for heavy metal accumulation. Nevertheless, the phytoremediation potential of willow in highly contaminated areas, like mine tailings, has not received much attention [26,32], and crucial information regarding the physiology of willow in such harsh environments is still lacking.

The extensive genetic variability observed in numerous species and hybrids of *Salix* spp. [27] offers a plethora of resources for selective breeding aimed at augmenting the phytoremediation capacity. Numerous studies have reported that the phytoremediation capacity varies significantly depends on the willow species or clones [33–36]. The *Salix* genus in China has been identified to comprise approximately 275 species [37]. However, limited research has been conducted on numerous Chinese species and clones for phytoremediation purposes in recent decades [29,38–42]. Furthermore, using willow original cuttings for direct phytoremediation is a more practical way to clean up polluted soil. However, there is limited research on the use of unrooted cuttings from different willow species/clones to thoroughly assess their potential for phytoremediation in contaminated soils [43], including mine tailings. As such, the focus of this study was to investigate the feasibility of utilizing unrooted cuttings in tailings with extremely high concentrations of heavy metals. The willow species *Salix jiangsuensis* '172', *S. integra*, *S. matsudana*, and *S. chaenomeloides*, which are indigenous to China, have been extensively cultivated throughout the country. *S. nigra*, native to the United States, has been demonstrated to have a specific capacity for heavy metal accumulation [34]. Therefore, the cuttings of these tree species were planted in two severely polluted mining tailings in this study. The survival ratio, growth performance, and metal uptake of these species were evaluated. The additional objective of this study was to elucidate the variation in the heavy metal tolerance and accumulation capacity among the selected willow clones for growth on mine tailings, as well as their potential utilization in phytoremediation of mine tailings in China. These results will be extremely helpful in selecting appropriate *Salix* species or clones for phytoremediation initiatives on mine tailings.

## 2. Materials and Methods

### 2.1. Site Description and Tailing Materials

To better replicate field conditions, we collected samples of Pb/Zn mine tailings from Fuyang, Hangzhou, China (30°07′38″ N, 119°50′39″ E) and Shaoxing, China (29°53′90″ N, 120°37′12″ E). Additionally, river sands were obtained from Fuyang, Hangzhou, China (30°03′43″ N, 119°57′12″ E). The physicochemical properties of the mine tailings and river sand were analyzed (Table 1), revealing that the concentrations of heavy metals in the mine tailings exceeded the risk screening values for soil contamination of agricultural land in China (GB 15618-2018) [44]. The single pollution index method is a dimensionless indicator used to assess the extent of soil contamination that can effectively reflect the excessive levels of multiple pollutants and the severity of pollution. The Nemerow index is a widely used comprehensive pollution index method that assesses the extent of heavy metal contamination in soils across an entire region. Detailed information can be found in the Supplementary Materials (Methods S1 and S2). The single pollution index (*Pi*) indicated a severe level of pollution for Cd, Pb, and Zn in the Pb/Zn tailings, as their *Pi* values significantly exceed 3 (Table S1). The *Pi* value of the Cu tailing was within the range

of 1–2, thus suggesting mild pollution. In the case of the Cu tailings, serious pollution is indicated by *Pi* values > 3 for Cu and Cd, while moderate pollution is indicated by a *Pi* value ranging from 2–3 for Zn. Furthermore, the *Pi* value for Pb was less than 1. Additionally, the Nemerow index ($P_N$) was calculated, revealing a significant co-pollution between Pb/Zn and Cu tailings (Table S1). The $P_N$ value for the Pb/Zn tailings was measured at 259.52, greatly surpassing the heavy pollution reference threshold of 3.

**Table 1.** Characteristics of three types of medium.

|  | River Sand | Pb/Zn Tailing | Cu Tailing |
|---|---|---|---|
| Pb (mg kg$^{-1}$) | 238 | 5850 | 108 |
| Zn (mg kg$^{-1}$) | 370 | 9980 | 1440 |
| Cu (mg kg$^{-1}$) | 11 | 186 | 1450 |
| CD (mg kg$^{-1}$) | 0.85 | 51.2 | 6.2 |
| Hydrolysable N (mg kg$^{-1}$) | 2 | 1 | 6 |
| Available P(mg kg$^{-1}$) | 17.4 | 11.1 | 10.9 |
| Rapidly available K (mg kg$^{-1}$) | 34.9 | 36.4 | 52.1 |
| Organic matter (g kg$^{-1}$) | 0.15 | 4.29 | 1.38 |
| pH | 8.12 | 7.59 | 7.92 |

### 2.2. Plant Materials

In this study, a total of fourteen *Salix* clones were selected, including the *S. jiangsuensis* '172', which is a hybrid of (*S. matsudana* × *S. chosenia arbutifolia*) × *S. matsudana* (Table 2). The capacity of these clones to produce a sizable amount of biomass during the two-year field experiment was the primary criterion in this specific selection.

**Table 2.** The *Salix* clones studied in the experiment.

| Clone ID | Species/Hybrid | Abbreviation | Origin |
|---|---|---|---|
| 172 | *Salix jiangsuensis* | SJ172 | Jiangsu provinces |
| 72 | *Salix nigra* | SN72 | United States |
| HY | *Salix integra* | SIHY | Shandong provinces |
| CT | *Salix integra* | SICT | Shandong provinces |
| WSH | *Salix integra* | SIWSH | Shandong provinces |
| YZB | *Salix integra* | SIYZB | Shandong provinces |
| 1 | *Salix matsudana* | SM1 | Zhejiang provinces |
| 14 | *Salix matsudana* | SM14 | Zhejiang provinces |
| 19 | *Salix matsudana* | SM19 | Zhejiang provinces |
| 34 | *Salix matsudana* | SM34 | Zhejiang provinces |
| 2 | *Salix chaenomeloides* | SC2 | Zhejiang provinces |
| 3 | *Salix chaenomeloides* | SC3 | Zhejiang provinces |
| 4 | *Salix chaenomeloides* | SC4 | Zhejiang provinces |
| 5 | *Salix chaenomeloides* | SC5 | Zhejiang provinces |

### 2.3. Experiment Methods

The experiment was conducted in a greenhouse of the Research Institute of Subtropical Forestry, Chinese Academy of Forestry, Hangzhou, China. In August 2019, healthy clone stem cuttings of uniform length (8 cm) and diameter (0.8 cm) were collected and rooted in black cylindrical plastic pots with dimensions of 30 cm in diameter and 18 cm in height. Each pot contained 7 cuttings of each clone, with the clones grown in river sand (Control, T0), Pb/Zn tailings (T1), and Cu tailings (T2), resulting in a total of 42 treatment units. These 42 treatment units were considered as one experimental block, and a total of four experimental blocks were arranged using a randomized block design. During a fifteen-month period, the cuttings were cultivated under natural lighting conditions, with day/night temperatures ranging from 25 to 35 °C and relative humidity levels maintained between 70% and 85%. Tensiometer measurements were used to regulate irrigation scheduling

throughout the growth phase to keep the soil moisture content roughly at field capacity. To sustain their growth, a weekly application of 0.2% ammonium nitrate was administered from June to August 2020. Any phytotoxic symptoms related to metal exposure were documented throughout the entire study period. Upon completion of the experiment, the survival rate of the cuttings and chlorophyll content in the leaves were quantified. Subsequently, all cuttings were harvested to assess their biomass, as well as their heavy metal and nutrient concentrations in the plant tissues.

$$\text{Survival rate (\%)} = \text{number of viable cuttings/total number of cuttings} \times 100\%.$$

### 2.4. Biomass Measurements

After harvesting, all samples were thoroughly rinsed with distilled water. Subsequently, the aboveground (leaf and stem) and belowground (cutting and root) portions were excised and separated. The belowground parts were immersed in 20 mmol/L $Na_2$–EDTA (Sinopharm Chemical Reagent Co., Ltd., Shanghai, China) for 15 min to remove metals. Following drying at 80 °C for 72 h, the dry biomass of the samples was measured using electronic scales (ME1002E, Mettler Toledo Instruments Ltd., Shanghai, China).

### 2.5. Estimation of Chlorophyll Content

The chlorophyll content of mature leaves (third–fourth leaf from the apex) in all treatments was estimated non-destructively using the Opti-Sciences CCM-200 chlorophyll content meter (Opti-Sciences Inc., Hudson, NH, USA), and the chlorophyll concentration index (CCI) was recorded.

### 2.6. Plant Sampling and Chemical Analysis

Following the pulverization of dried plant samples into a fine powder, 0.2 g of each sample was digested using a solution of 4 mL $HNO_3$ and 1 mL $HClO_4$ (Sinopharm Chemical Reagent Co., Ltd., Shanghai, China). The concentrations of Pb, Zn, Cu, and Cd in the dried plant samples were determined using inductively coupled plasma–mass spectrometry (ICP-MS, NexION300D, PerkinElmer Inc., Shelton, CT, USA). To ensure the precision of analyses, certified reference materials, namely mixed shrub shoots from Pb/Zn mine tailings (GBW 07602, National Research Center for Certified Reference Materials, Beijing, China), were employed. Good agreement was found between the certified values and our technique (Table S2). Bioconcentration factor (BCF) values at the end of the experiment were calculated using BCF = $A_{tissues}/A_{soil}$ [45], where $A_{tissues}$ (mg kg$^{-1}$) represents the total accumulation of heavy metals in tissues, and $A_{soil}$ (mg kg$^{-1}$) denotes the concentration of heavy metals in the tailings. Translocation factor (TF) values were calculated using TF = $A_a/A_b$ [45], where $A_a$ (mg kg$^{-1}$) indicates the total accumulation of heavy metals in the aboveground parts, and $A_b$ (mg kg$^{-1}$) represents the total accumulation of heavy metals in the belowground parts.

### 2.7. Tailing Property Measurements

After air-drying, the tailing samples were sieved through a <2 mm mesh size to achieve a homogenous mixture. Subsequently, the samples underwent extraction with 5 mL of combined acids (65% $HNO_3$ and 70% $HClO_4$), followed by analysis of the total metals (Pb, Zn, Cd, and Cu) using ICP-MS. The dichromate heating oxidation method was used to measure the organic matter in the soil [46]. Using the alkaline diffusion approach, soil hydrolytic N was determined [46]. A colorimetry technique was employed to measure the soil available P after $NaHCO_3$ extraction (Sinopharm Chemical Reagent Co., Ltd., Shanghai, China) [46]. Flame photometry detection (AAS, ICE3500, Thermo Scientific, Waltham, MA, USA) was conducted to determine soil available K following the extraction with 1 M $NH_4OAc$ (Sinopharm Chemical Reagent Co., Ltd., Shanghai, China) [46]. The pH was determined using an FE32 pH meter (Mettler Toledo Instruments Ltd., Shanghai, China) with a 1:2.5 ratio (*m:v*) of tailings to distilled water.

*2.8. Statistical Analysis*

Two comprehensive evaluation methods were applied to assess the phytoremediation of willow clones for the two mining tailings. The TOPSIS (technique for order preference by similarity to ideal solution) method is a method of ranking a limited number of evaluation objects according to their proximity to the idealized target, which is to evaluate the relative advantages and disadvantages of the existing objects. The comprehensive bioaccumulation index (CBAI) was proposed as an integrated evaluation approach, combining plant biomass and heavy metal concentrations [47]. Further details can be found in the Supplementary Materials (Methods S3 and S4).

The statistical analyses were conducted using R 4.3.2. Two-way analysis of variance was performed for all variables, with treatment and clones/species as the different factors. The least significant difference test was used to compare the means post hoc at a significance level of 0.05. Shapiro's test and Levene's test were used to evaluate the homoscedasticity and normal distribution assumptions prior to the significance tests, respectively. In cases where these assumptions were violated, logarithmic transformations were applied to ensure a normal distribution. The results were standardized and subsequently analyzed using OriginPro 2022 for principal component analysis (PCA). Data are presented as mean ± standard error, with at least three replicates.

## 3. Results

*3.1. Cutting Survival*

The survival rate of the cuttings in the control group was 100% for all clones, whereas for cuttings grown in tailings, it was significantly lower than the former ($p < 0.05$). Although the different tailing treatments did not have a significant effect on the survival of cuttings in this study, individual cuttings grown in the Pb/Zn tailings exhibited an average survival rate of 28.93%, while those grown in Cu tailings demonstrated a slightly higher survival rate of 31.98%. In addition, there was a significant difference in survival rates among the clones ($p < 0.05$, Table S3). *S. jiangsuensis* '172' exhibited the highest survival rate, with 71.40% and 85.70% of the cuttings surviving in the Pb/Zn and Cu tailings, respectively. However, *S. chaenomeloides* '5' had the lowest survival rates, with only 14.30% in both the Pb/Zn and Cu tailings (Figure 1). Overall, *S. chaenomeloides* demonstrated the lowest survival rates in both types of tailings, with only 19.07% and 22.64% survival observed for the Pb/Zn and Cu tailings, respectively.

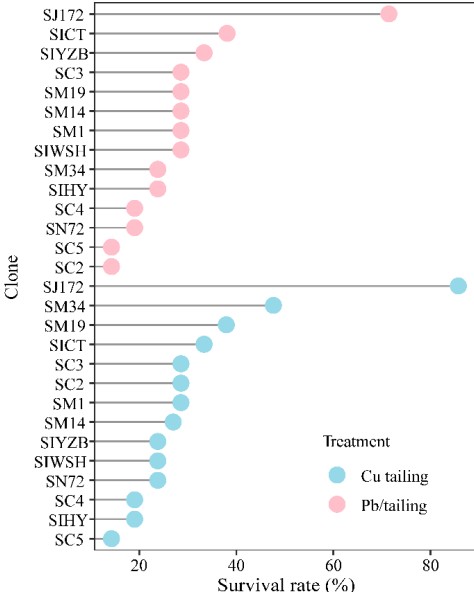

**Figure 1.** Cutting survival of 14 *Salix* clones in the Pb/Zn and Cu tailings after 15 months.

### 3.2. Biomass

Following the experiment, it was found that both the treatment and the clone had a considerable impact on the biomass of plants (Table S3). Compared to clones produced in river sand, those grown in tailings typically had a lower aboveground biomass (Figure 2). The aboveground biomass of clones growing in the Pb/Zn and Cu tailings was 15.9% and 12.9% lower than that of the control group, respectively. However, the aboveground biomass of *S. matsudana* '1' and '14', as well as *S. chaenomeloides* '3' and '4', exhibited a biomass increase in the Pb/Zn tailings, while a similar trend was observed for *S. integra* 'HY' and *S. matsudana* '14' and '19' in the Cu tailings (Figure 2 and Table S4). Meanwhile, the cuttings grown in the tailings showed varying aboveground biomass yields. *S. nigra* '72', *S. integra* (except 'CT'), *S. matsudana* '19' and '34', and *S. chaenomeloides* '2' in particular showed increased aboveground biomass production in the Cu tailings compared to the Pb/Zn tailings (Figure 2 and Table S4). As a group, *S. chaenomeloides* showed a greater sensitivity to the Cu tailings, while *S. nigra* and *S. integra* were more susceptible to the Pb/Zn tailings. The aboveground biomass within the same tailings showed statistically significant differences among the clones ($p < 0.05$). The highest aboveground biomass was observed in *S. chaenomeloides* '4' (Pb/Zn tailings) and *S. matsudana* '19' (Cu tailings). Overall, *S. chaenomeloides* exhibited the highest aboveground biomass in the control and Pb/Zn tailings groups, whereas *S. matsudana* showed the greatest aboveground biomass in the Cu tailings.

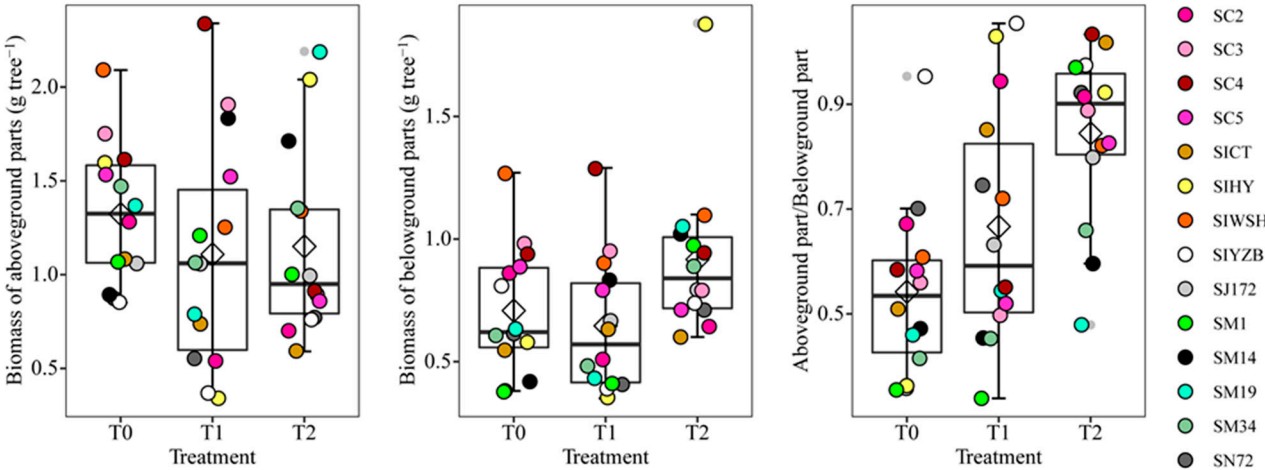

**Figure 2.** Cutting biomass (g tree$^{-1}$) of 14 *Salix* clones in different treatments after 15 months.

The results indicated that the mean belowground biomass of the cuttings in the control group was 0.71 g tree$^{-1}$ (Figure 2). A significant increase in the belowground biomass was observed when the cuttings were cultivated in the Cu tailings compared to the control ($p < 0.05$), while a slight reduction was noted in the Pb/Zn tailings (Figure 2). Between the two tailings, the cuttings' biomass production differed noticeably. Particularly, when growing in the Cu tailings, as opposed to the Pb/Zn tailings, *S. jiangsuensis* '172', *S. nigra* '72', *S. integra* (except 'CT'), *S. matsudana*, and *S. chaenomeloides* '2' produced greater belowground biomass (Figure 2 and Table S4). Furthermore, a significant variation was observed among the clones within the same tailings ($p < 0.05$). Typically, *S. chaenomeloides* exhibited a higher belowground biomass than the other species in the Pb/Zn tailings, with *S. chaenomeloides* '4' exhibiting the highest belowground biomass (Table S4). In the Cu tailings, *S. integra* demonstrated a higher belowground biomass than the other species, with *S. integra* 'HY' producing significantly greater amounts of belowground biomass (1.88 g tree$^{-1}$) than its counterparts (Table S4).

A significant difference was seen among treatments based on the belowground/aboveground ratio results ($p < 0.05$, Table S3). The mean values of the belowground/aboveground ratio ranged from 0.35 to 0.99 (control), from 0.37 to 1.03 (Pb/Zn tailings),

and from 0.48 to 1.11 (Cu tailings). Compared with the control, the mean values for the belowground/aboveground ratio increased significantly in the tailing treatments, especially in the Cu tailings (0.85). Statistically significant differences were observed among the clones in terms of belowground/aboveground ratio values within the same tailings.

### 3.3. Chlorophyll Concentration Index

The CCI values ranged from 16.13 to 23.93 (control), from 8.90 to 20.16 (Pb/Zn tailings), and from 11.84 to 20.08 (Cu tailings) (Figure 3). Statistically significant differences were also observed between the treatments and clones in terms of CCI values ($p < 0.05$, Table S3). However, no statistically significant differences were found among all treatments for *S. jiangsuensis* '172', *S. integra* 'HY' and 'CT', *S. matsudana* '1' and '19', and *S. chaenomeloides* '2' (Table S5). Compared to other species, *S. chaenomeloides* generally showed a higher CCI value in the Pb/Zn and control tailings, while *S. jiangsuensis* '172' displayed the greatest value in the Cu tailings. *S. nigra* '72', on the other hand, consistently exhibited lower CCI values in all treatments. These findings imply that *S. nigra* '72' responded poorly to the environmental changes and was less robust than the native species.

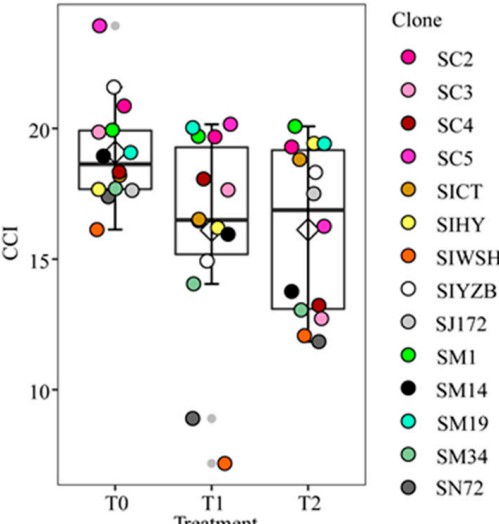

**Figure 3.** Chlorophyll concentration index (CCI) of leaves of 14 *Salix* clones in different treatments after 15 months.

### 3.4. Accumulation and Translocation of Metals in the Cuttings

In principle, the concentrations of all the heavy metals were found to be higher in the belowground parts compared to the aboveground parts of the cuttings cultivated in the tailing treatments (Figure 4). Zn was found at higher quantities in the tissues of the cuttings growing in the tailings than Pb, Cu, and Cd. The heavy metal concentrations in the belowground parts were Zn > Pb > Cu > Cd (Pb/Zn tailings) and Zn > Cu > Pb > Cd (Cu tailings) in that order (Figure 4). However, for the aboveground parts, the order was Zn > Cu > Pb > Cd (Pb/Zn tailings) and Zn > Cu > Cd > Pb (Cu tailings) (Figure 4). Additionally, the results showed that the four metals concentrations in all the cuttings' tissues varied significantly depending on the treatments and clones ($p < 0.05$, Table S3). The concentrations of Pb, Zn, and Cd in the clones cultivated in the Pb/Zn tailings were generally higher compared to those cultivated in the Cu tailings (Figure 4).

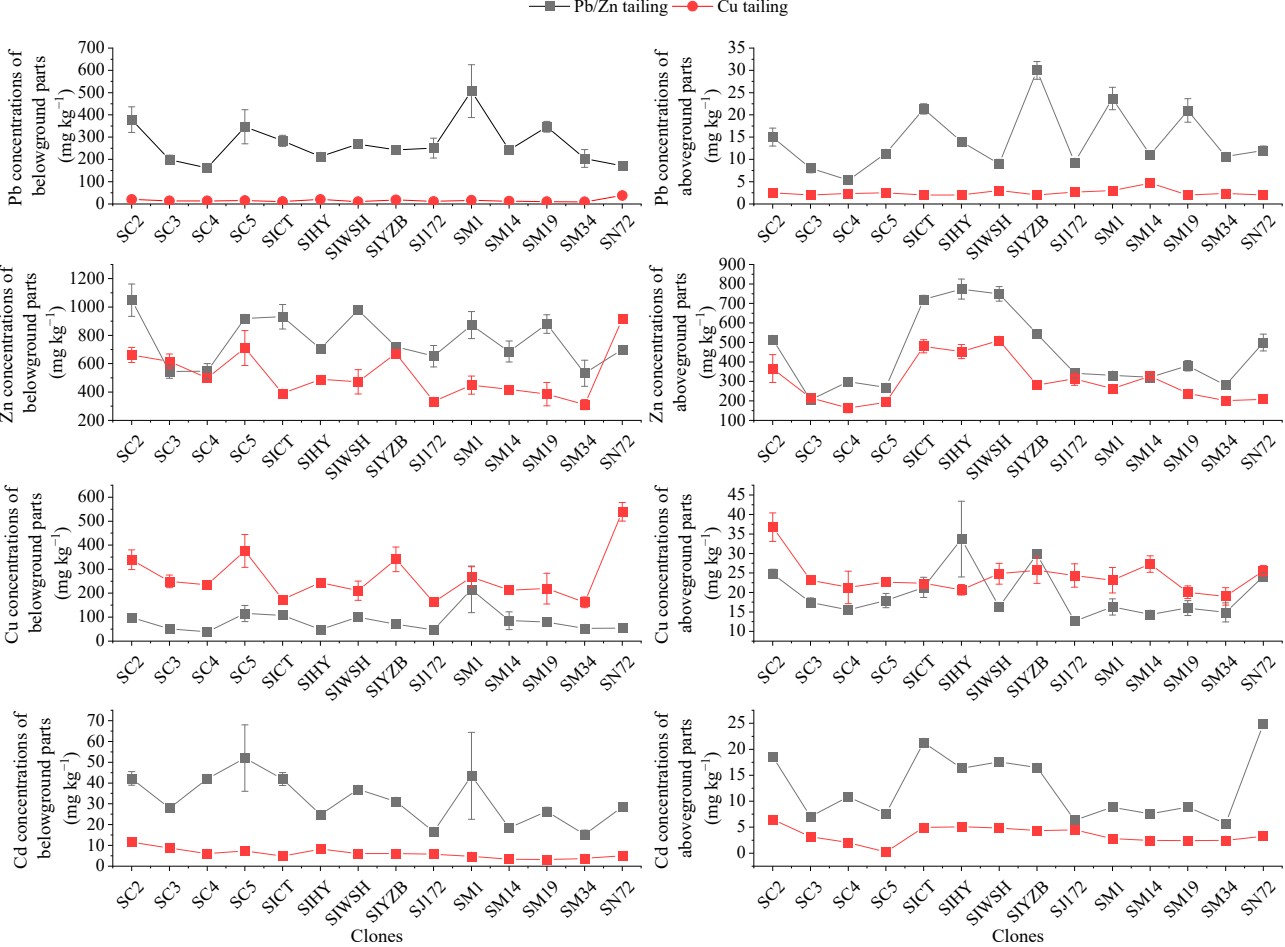

**Figure 4.** Heavy metal concentrations (mg kg$^{-1}$) in aboveground and belowground parts of 14 *Salix* clones following different treatments after 15 months.

Pb concentrations in the aboveground parts ranged from 5.3 to 30.0 mg kg$^{-1}$ (Pb/Zn tailings) and 2.0 to 4.7 mg kg$^{-1}$ (Cu tailings) across the 14 clones. As shown in Figure 4, the maximum mean Pb concentrations were observed in *S. integra* 'YZB' (30.0 mg kg$^{-1}$, Pb/Zn tailings) and *S. matsudana* '14' (4.7 mg kg$^{-1}$, Cu tailings). In contrast, the belowground parts exhibited a range of Pb concentrations from 161.7 to 506.7 mg kg$^{-1}$ (Pb/Zn tailings) and from 9.3 to 37.3 mg kg$^{-1}$ (Cu tailings). In the belowground parts, *S. matsudana* '1' and *S. nigra* '72' exhibited a greater ability for Pb accumulation in the Pb/Zn and Cu tailings, respectively. Generally, *S. chaenomeloides* '4' grown in the Pb/Zn tailings showed the lowest amounts of Pb in both the above- and belowground parts.

Figure 4 shows that the concentrations of Zn in the samples ranged from 205 to 1048 mg kg$^{-1}$ (Pb/Zn tailings) and from 163 to 914 mg kg$^{-1}$ (Cu tailings). Generally, *S. integra* could accumulate much more Zn in their tissues in both types of tailings, except for the belowground parts in the Cu tailings. Notably, the highest concentration in the belowground parts was recorded in *S. chaenomeloides* '2' in the Pb/Zn tailings and *S. nigra* '72' in the Cu tailings.

The average Cu concentrations ranged from 12.6 to 33.7 mg kg$^{-1}$ (aboveground, Pb/Zn tailings), from 18.9 to 36.8 mg kg$^{-1}$ (aboveground, Cu tailings), from 38.6 to 114.9 mg kg$^{-1}$ (belowground, Pb/Zn tailings), and from 161.7 to 539.3 mg kg$^{-1}$ (belowground, Cu tailings) (Figure 4). *S. jiangsuensis* '172' and *S. matsudana* '34' exhibited a relatively low capacity for Cu accumulation in their tissues, while *S. integra* 'HY' (Pb/Zn tailings) and *S. chaenomeloides* '2' (Cu tailings) demonstrated higher levels of Cu concentrations in their aboveground

parts; the highest Cu concentrations in the belowground parts were noted in *S. matsudana* '1' (Pb/Zn tailings) and *S. nigra* '72' (Cu tailings).

The average concentrations of Cd in the aboveground parts were 12.7 mg kg$^{-1}$ and 3.5 mg kg$^{-1}$ for all clones grown in the Pb/Zn and Cu tailings, respectively (Figure 4). Based on the results, the lowest Cd concentrations in the aboveground parts were found in *S. matsudana* '34' (5.7 mg kg$^{-1}$, Pb/Zn tailing) and *S. chaenomeloides* '5' (0.23 mg kg$^{-1}$, Cu tailings). Meanwhile, the lowest Cd concentrations in the belowground parts were observed in *S. matsudana* '34' (15.2 mg kg$^{-1}$, Pb/Zn tailings) and *S. matsudana* '19' (3.1 mg kg$^{-1}$, Cu tailings). Generally, *S. matsudana* had the least efficient accumulation of Cd in the Cu tailings. In the Cu tailing, *S. chaenomeloides* '2' concentrated most Cd in its tissues. Moreover, *S. nigra* '72' and *S. chaenomeloides* '5' exhibited the highest Cd concentrations in their aboveground and belowground parts, with an average concentration of 24.9 and 52.0 mg kg$^{-1}$, respectively, in the Pb/Zn tailing.

The BCF values for the samples are presented in Figure 5. The results indicated that all four metals' BCF values were less than 1, and the clones had a greater affinity for Cd enrichment but a lower affinity for Pb enrichment. The findings show that these values are significantly impacted by the type of treatment ($p < 0.05$, Table S3). The average heavy metal BCF values in the Cu tailings were higher than those in the Pb/Zn tailings, with the exception of Cu. The BCF values of the cuttings grown in the same tailing were significantly influenced by the clones as well. Specifically, *S. chaenomeloides* '2' and *S. integra* 'CT' exhibited a higher accumulation of heavy metals in their organs within the Pb/Zn tailing; while *S. matsudana* '34' demonstrated lower efficiency in metal accumulation. Among all the clones, except for Cd, *S. nigra* '72' had the greatest BCF values for metals when grown in the Cu tailings, while *S. matsudana* '19' and *S. matsudana* '34' showed lower accumulation of heavy metals in their organs.

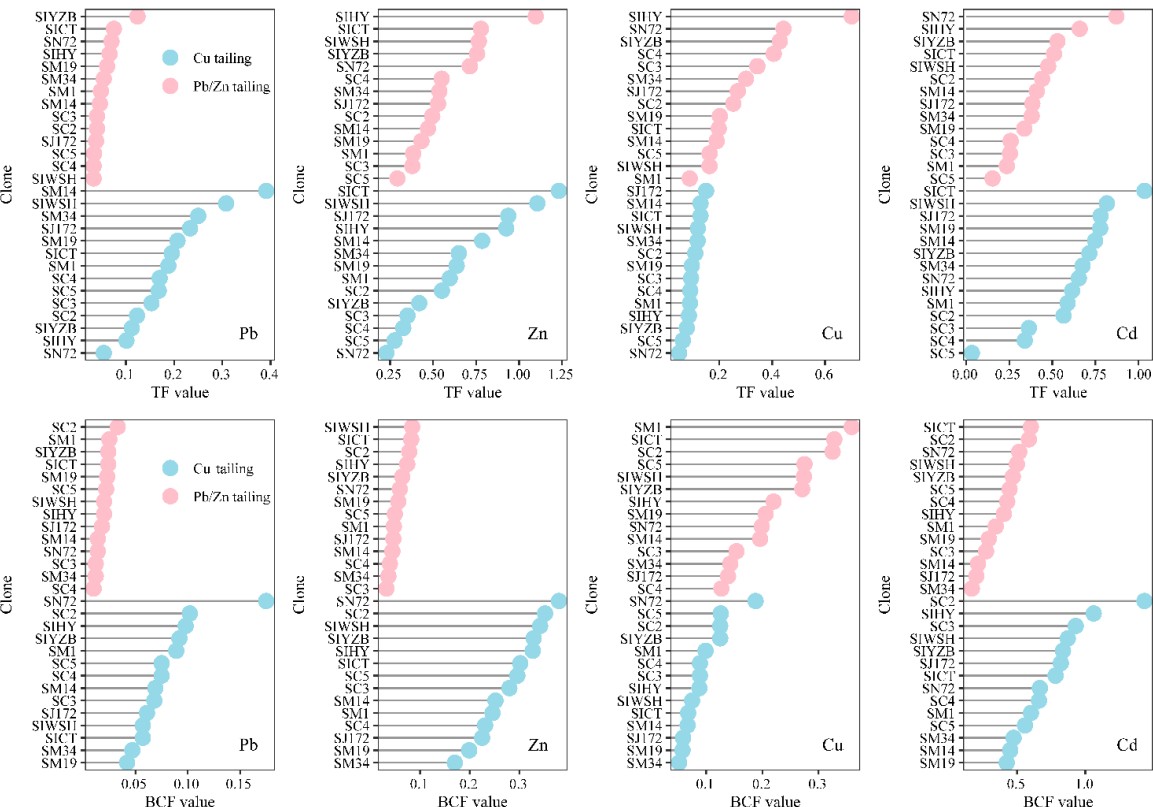

**Figure 5.** Bioconcentration factor (BCF) and translocation factor (TF) of Pb, Cu, Zn, and Cd of 14 *Salix* clones in the Pb/Zn and Cu tailings.

Generally, the average TF values for Pb in the cuttings were below 0.2, especially in the Pb/Zn tailings, where they were only 0.05. However, the cuttings exhibited relatively higher TF values for Zn and Cd, especially in the Cu tailings (Figure 5). Furthermore, with the exception of Cu, the TF values observed in cuttings grown in the Cu tailings were higher than those found in the Pb/Zn tailings. Significant variations were noted among the 14 clones regarding their TF values for all metals present within a given type of tailing. The lowest TF values for Pb, Zn, and Cu were observed in *S. nigra* '72' grown in the Cu tailings. Meanwhile, it was found that *S. chaenomeloides* '5' exhibited poor translocation ability for Zn and Cd from the belowground parts to the aboveground part in both tailings. As an exception, *S. integra* 'CT' exhibited TF values greater than 1.0 and demonstrated the highest TF values for Zn (1.232) and Cd (1.035) among the clones grown in the Cu tailing. *S. integra* 'HY' and *S. integra* 'WSH' also displayed higher Zn translocate efficiencies, with mean TF values of 1.097 ('HY') and 1.106 ('WSH') in the Pb/Zn and Cu tailings, respectively.

### 3.5. PCA Analysis and Evaluation of Phytoremediation Potential

The total variance can be explained by three principal components, accounting for 85.33% of the variance. The first component, representing 47.65% of the total variance, exhibited a positive correlation with the concentrations of Pb, Zn, and Cd and was negatively associated with survival rate. PC2 accounted for 21.46% of the total variance and was positively associated with biomass and negatively linked with the aboveground concentration of Cu (Figure 6). These results suggested that the survival rate and biomass of the plants was reduced with an increase in the Zn, Pb, Cd, and Cu concentrations in the tissues under the tailing conditions. The principal components clearly highlight the clones treated with different tailings. The growth and heavy metal accumulation data for the Pb/Zn tailings exhibit greater variability compared to those for the Cu tailings.

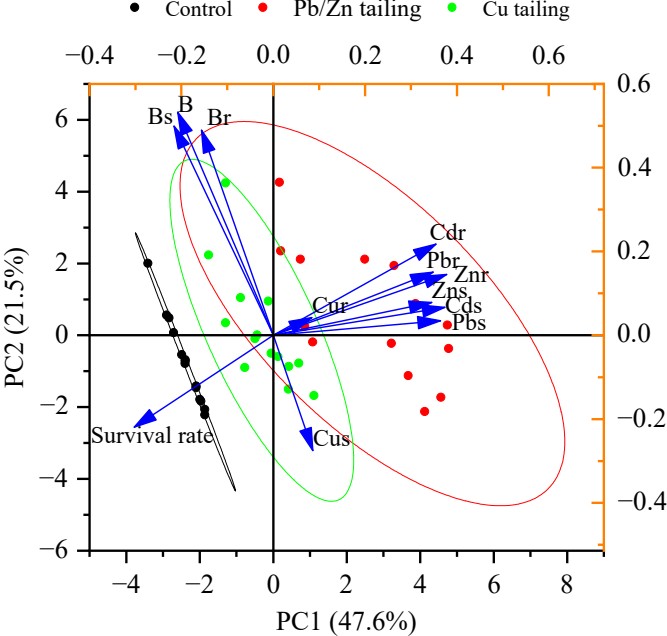

**Figure 6.** Principal component analysis of growth and heavy metal concentration under different tailing treatments. Bs: Biomass of aboveground; Br: Biomass of belowground; B: Biomass of all plant; Pbs: Pb concentration of aboveground; Zns: Zn concentration of aboveground; Cus: Cu concentration of aboveground; Cds: Cd concentration of aboveground; Pbr: Pb concentration of belowground; Znr: Zn concentration of belowground; Cur: Cu concentration of belowground; Cdr: Cd concentration of belowground.

*S. integra* 'WSH' had the highest values (0.862) in the Pb/Zn tailings of any family, according to the CBAI index, while *S. chaenomeloides* '4' and '5' also showed somewhat

higher values than the other families. In the Cu tailings, *S. integra* 'HY' had the highest CBAI values (Table 3). According to the TOPSIS method, *S. jiangsuensis* '172' had the largest C$i$ (the degree of proximity) values in both tailings. Additionally, *S. integra* 'WSH', *S. integra* 'CT', and *S. matsudana* '1' also had large C$i$ values that ranged from 0.443 to 0.496 in the Pb/Zn tailings. Similarly, the C$i$ values of *S. integra* 'HY' were high in the Cu tailings (Table 3).

**Table 3.** Evaluation of the phytoremediation potential of different clones.

| Clones | TOPSIS | | | | CBAI | | | |
|---|---|---|---|---|---|---|---|---|
| | Pb/Zn Mine Tailing | | Cu Mine Tailing | | Pb/Zn Mine Tailing | | Cu Mine Tailing | |
| | Score | Rank | Score | Rank | Score | Rank | Score | Rank |
| *Salix jiangsuensis* 172 | 0.4399 | 3 | 0.5004 | 1 | 0.2460 | 9 | 0.1034 | 12 |
| *Salix nigra* 72 | 0.2896 | 13 | 0.4054 | 4 | 0.0866 | 11 | 0.3947 | 2 |
| *Salix integra* HY | 0.3113 | 11 | 0.4518 | 2 | 0.0121 | 14 | 1.0000 | 1 |
| *Salix integra* CT | 0.4638 | 2 | 0.3009 | 10 | 0.5356 | 5 | 0.0000 | 14 |
| *Salix integra* WSH | 0.4173 | 4 | 0.3541 | 5 | 0.8622 | 1 | 0.3721 | 3 |
| *Salix integra* YZB | 0.3854 | 6 | 0.3027 | 9 | 0.0805 | 13 | 0.2180 | 8 |
| *Salix matsudana* 1 | 0.4725 | 1 | 0.2634 | 12 | 0.5310 | 6 | 0.2458 | 6 |
| *Salix matsudana* 14 | 0.3677 | 7 | 0.3518 | 6 | 0.5535 | 4 | 0.2978 | 4 |
| *Salix matsudana* 19 | 0.3023 | 12 | 0.3461 | 7 | 0.2163 | 10 | 0.2667 | 5 |
| *Salix matsudana* 34 | 0.1806 | 14 | 0.3070 | 8 | 0.0859 | 12 | 0.0791 | 13 |
| *Salix chaenomeloides* 2 | 0.3394 | 10 | 0.4254 | 3 | 0.3796 | 8 | 0.2435 | 7 |
| *Salix chaenomeloides* 3 | 0.3463 | 9 | 0.2808 | 11 | 0.4915 | 7 | 0.1811 | 9 |
| *Salix chaenomeloides* 4 | 0.4017 | 5 | 0.1904 | 14 | 0.7794 | 2 | 0.1758 | 10 |
| *Salix chaenomeloides* 5 | 0.3625 | 8 | 0.2321 | 13 | 0.7832 | 3 | 0.1719 | 11 |

## 4. Discussion

### 4.1. Variation in the Survival Rate and Biomass Production

Numerous studies have consistently demonstrated that phytoremediation is not suitable for highly contaminated sites, such as mining areas, due to the higher concentrations of heavy metals that can influence the survival rates and biomass yield [43]. Unsurprisingly, cuttings cultivated in the tailings had significantly lower survival rates, with most clones failing to exceed 40%. These findings demonstrate a discrepancy compared to field experiment results [43]. A possible reason for this phenomenon could be that the concentrations of heavy metals in our study's mine tailings were significantly higher than those in their experiment. Furthermore, the growth and tolerance capacity of the plants may also be influenced by the diameter of the initial cuttings [48]. In our experiment, the presence of fine cuttings with a diameter less than 1 cm could potentially impact the survival rates of the cuttings in both tailings. However, surprisingly, the survival rates of *S. jiangsuensis* '172' exceeded 70%, aligning with our initial predictions due to its manifestation of indigenous traits and substantial heterotic effects, where hybrids typically outperform their parents, particularly in terms of growth potential and yield [36]. This may also facilitate the initiation of plant tolerance mechanisms by enhancing plant antioxidant activity [49]. Many studies have also reported that *S. jiangsuensis* '172' was resistant to the presence of pollutants in a pot experiment [41]. Castiglione et al. [50] suggested that in heavily contaminated soil, when the average plant survival rate reaches approximately 30%, it can be deemed promising for accomplishing the actual objective. In brief, these findings suggest that *S. jiangsuensis* '172' has a high tolerance capacity and would be a recommended candidate for future phytoremediation projects in tailings with extremely high concentrations of heavy metals. The other clones could be considered candidates for phytoremediation purposes in lower and moderately contaminated environments.

In general, plant biomass serves as a reliable indicator for assessing the growth performance of plants subjected to heavy metal stress [33]. In this study, the cuttings could grow in the tailing treatments; however, a reduction in biomass was observed, except for

belowground biomass in the Cu tailings. These symptoms are in line with other research studies [26,36,51]. In this experiment, the heavy metal concentrations in both the tailings exceeded the toxic threshold concentration for plants in soil [52]. Moreover, the concentrations of heavy metals in the cuttings significantly surpassed the phytotoxic threshold concentrations for plants [53]. In our research, the aboveground part concentrations of Pb (Pearson r = −0.338, *p* < 0.05, N = 42), Zn (Pearson r = −0.356, *p* < 0.05, N = 42), Cu (Pearson r = −0.473, *p* < 0.01, N = 42), and Cd (Pearson r = −0.356, *p* < 0.05, N = 42) showed a strong negative correlation with the biomass of the aboveground part. The biomass of the belowground part also showed a significantly negative correlation with the concentration of Pb in the aboveground part (Pearson r = −0.373, *p* < 0.05, N = 42). These findings could potentially explain the reduced biomass production in the mine tailings. However, some clones produced more biomass when grown in the tailings compared to in the control group, suggesting that the surviving cuttings have efficient mechanisms to tolerate heavy metals under the present experimental conditions. For example, most of the tested cuttings allocated more resources to the belowground parts under stress conditions in this study, and this modification served as a tolerance mechanism when grown in the tailings. On the other hand, the 100% survival rate led to high plant density in the control group, resulting in biomass production that was relatively smaller for some clones.

The current study demonstrated large variations in the biomass performances among the species and clones, which corroborates with most previous findings on *Salix* clones cultivated in contaminated environments [18,36,38,54]. In this study, the highest aboveground biomass (2.34 g tree$^{-1}$) and belowground biomass (1.99 g tree$^{-1}$) in the Pb/Zn tailings were produced by *S. chaenomeloides* '4'; these were approximately seven and four times higher than the lowest aboveground and belowground biomass found in *S. integra* 'HY', respectively. However, *S. chaenomeloides* '4' only exhibited 50% biomass production in the Cu tailings compared to the Pb/Zn tailings, whereas the biomass of *S. integra* 'HY' significantly surpassed that of the other clones. The observed phenomenon may be due to considerable heterogeneity in genotypic sensitivity to various heavy metals and tolerance to excessive heavy metals, which is also influenced by environmental factors. In general, most clones exhibited higher biomass production in the Cu tailings, which could be attributed to the comparatively lower pollution levels and higher nitrogen content of the Cu tailings compared to the Pb/Zn tailings. However, *S. chaenomeloides* '3', '4', and '5' demonstrated significantly smaller biomass in the Cu tailings than in the Pb/Zn tailings, suggesting their heightened sensitivity towards Cu. Notably, *S. jiangsuensis* '172' was unable to generate a significant amount of biomass across the clones despite having very high survival rates. Weih [55] highlighted that selecting plants based solely on biomass is an insufficient criterion for phytoremediation projects, as it may be associated with inadequate resistance to contaminants. Thus, *S. jiangsuensis* '172' should be recommended for phytoremediation in the southern regions of China.

*4.2. Heavy Metals Accumulation in Willow Tissues*

Numerous research studies have reported that willow clones possess high heavy metal accumulation potentials [21,26,56–58]. In comparison with data from previous studies, the concentrations of heavy metals in most of the tested cuttings were either similar to or higher than those reported for other willow species [25,43,58]. One plausible explanation for this is the extremely high contamination in our test mine tailings. Many authors have emphasized that the data regarding the concentrations of heavy metals in plant tissues found in different studies are hard to compare because of the different plant species used, growth stages recorded, experimental conditions, and exposure durations, all of which have a significant impact on the uptake and accumulation of heavy metals in plant tissues [27,39]. Consequently, it is not unexpected that the concentrations of heavy metals in the current study were comparatively lower than those reported in previous hydroponic studies [38,40,59]. Despite the higher concentrations of heavy metals in the tailings compared to the control treatment, the majority of clones in our experiments

had relatively low BCF and TF values, suggesting that all cuttings in the current study could only uptake and transport a limited quantity of metals from the tailings to the plant tissues. The results of this study clearly demonstrate that the pH values in the Pb/Zn and Cu tailings exceeded 7, indicating alkaline conditions. Previous research studies have indicated that alkaline conditions may significantly limit the bioavailability of heavy metals in soil [60]. Mendez and Maier [61] suggested that nutrient deficiencies could also influence element availability in mining areas. In addition, the organic matter content in the tailings was found to be significantly higher than that in the control in this study. Research has demonstrated a negative correlation between the soil's available heavy metal concentration and its organic carbon content level [62]. These factors likely contribute to the relatively low BCF and TF values observed for the *Salix* cuttings in our study. Furthermore, the alkaline environment was found to be unsuitable for most plant development. Consequently, the growth of plant roots was hindered under these adverse conditions, leading to a significant impediment in the uptake of heavy metals by *Salix*. Additionally, it has been observed that metals can interact antagonistically or synergistically [63], resulting in complex interactions among different elements that can influence the accumulation and translocation of heavy metals in plants [63]. According to our findings, all clones had lower heavy metal transfer capabilities in the tailings with extremely high concentrations of heavy metals. This high retention indicated that most test *Salix* clones might adopt an excluder strategy for tolerance, which represents the important tolerance mechanism for phytoremediation processes [64]. The findings of this study also demonstrated that all clones could easily take up and translocate Zn and Cd from the soil to aboveground parts, while translocating Pb from the roots to the aboveground parts was limited. This tendency was noted by other researchers as well [26,65].

Consistent with our findings, previous research has also observed significant variations in heavy metal accumulation and distribution among *Salix* species or clones; this can be attributed to their genetic variability in terms of metal accumulation capacity [34,36]. For example, the concentrations of heavy metals in *S. chaenomeloides* '2' from the Pb/Zn tailings was significantly higher compared to that of *S. chaenomeloides* '3'. These findings revealed that genetic variables controlled the features of heavy metal accumulation in willow plants. Additionally, we also found that the concentration of heavy metals in the same clone varied significantly between the tailings. For instance, when cultivated with the Cu tailings, *S. nigra* '72' showed increased Zn accumulation in the roots. The aforementioned results indicated that environmental factors affected the concentrations of heavy metals in willow tissues.

*4.3. Potential for Phytoremediation*

The fact that *Salix* spp. is not a hyperaccumulator is widely acknowledged, making it best suited for use in mildly or moderately contaminated soils and rarely in highly contaminated sites. The BCF and TF values were used to assess a plant's capacity to accumulate metals from soils and translocate these metals from the roots to the shoot, respectively. The study also revealed that the tested willow exhibited low BCF and TF values, thereby indicating its unsuitability as a phytoremediation candidate for sites with severe pollution. However, the efficiency of phytoremediation can be influenced by other factors, such as plant survival rate, biomass production, and heavy metal concentration. Despite being important indicators of phytoremediation, BCF and TF ignore plant biomass and survival [21]. In this study, two comprehensive evaluation methods combining plant growth and heavy metal concentration were used to evaluate the phytoremediation potential of willow clones. According to the TOPSIS method, *S. jiangsuensis* '172' showed a higher *Ci* value (the degree of proximity), while its CBAI value was relatively low, indicating that it had a high survival rate and a relatively low capacity to accumulate heavy metals in its tissues in both tailings. Therefore, it is recommended that this species is considered as a viable option for mine tailing afforestation and ecological restoration. The future application of this clone could involve increasing biomass per unit area through dense planting to facilitate

the accumulation of heavy metals. Additionally, a short-cycle rotation (2 years) could be implemented to gradually remove heavy metals from the soil. In our pot experiment, *S. integra* accumulated relatively high concentrations of Zn and Cd in their aboveground parts. Meanwhile, the surviving cuttings of some clones could produce more biomass, such as *S. integra* 'WSH' (both tailings) and 'HY' (Cu tailing). However, lower survival rates were observed in this species. Considering the *Ci* and CBAI values, *S. integra* 'WSH' and 'HY' demonstrated their potential for phytoextraction in severely contaminated Pb/Zn and Cu tailings, respectively. This was consistent with what we found earlier [66]. The other clones should not be recommended for phytoremediation projects in highly contaminated areas, at least not until further research has been conducted to assess their suitability.

Compared to findings from previous studies [26,65], the average contents of heavy metals in all the clones were generally lower, particularly in the aboveground parts due to their limited biomass production. For instance, the average contents of aboveground parts in the Pb/Zn tailings were from 0.005 to 0.029 mg tree$^{-1}$ Pb, from 0.20 to 0.94 mg tree$^{-1}$ Zn, from 0.012 to 0.036 mg tree$^{-1}$ Cu, and from 0.006 to 0.022 mg tree$^{-1}$ Cd; the average contents in belowground parts were higher than those in the aboveground parts, except for Zn in some clones (Tables S6 and S7). In the current research, biomass production was reduced due to an imbalance of the nutrient elements, nutrient deficiency, and metal toxicity. At the later stages of the experiment, several symptoms of toxicity were alleviated through weekly applications of 0.2% ammonium nitrate spray. The diameter of the cuttings used in this study, which is less than 1 cm, may potentially account for the limited biomass observed in the tested plants. Previous research has indicated that thick cuttings (>1 cm diameter) exhibit a greater biomass yield compared to fine cuttings (<1 cm diameter), and there is a higher concentration of heavy metals in the aboveground section. Furthermore, thick cuttings show improved phytoextraction and rhizofiltration efficiency [48]. Therefore, it is recommended that thick cuttings (>1 cm diameter) should be used in future phytoremediation protocols for extremely high concentrations of heavy metals in the tailings. Additionally, regular spraying with 0.2% ammonium nitrate should be considered. Mleczek et al. [22] reported that *S. alba* could produce 17 t of fresh and 7 t of dry matter per ha annually, with a density of 0.25 plant per m$^2$. However, in heavily contaminated areas, the survival rate and biomass production of willow would be lower than those in a normal environment. Consequently, based on findings from this study and others, increasing the willow plant density to up to 30,000 cuttings per ha could ensure optimal biomass production. If the aboveground and belowground biomasses could reach approximately 25 and 10 g tree$^{-1}$, respectively, in the first growing season while maintaining constant heavy metal concentrations in their organs as described in this study, the heavy metal removal in the aboveground part would comprise 10.5 g Pb per ha, 300 g Zn per ha, 15 g Cu per ha, and 7.5 g Cd per ha in the Pb/Zn tailing. Simultaneously, 75 g Pb per ha, 210 g Zn per ha, 24 g Cu per ha, and 9 g Cd per ha were stored in the belowground part. In the Cu tailings, the amounts of extracted heavy metals were 1.9 g Pb per ha, 225 g Zn per ha, 18 g Cu per ha, and 2.6 g Cd per ha in the aboveground part, while they were 4.5 g Pb per ha, 150 g Zn per ha, 75 g Cu per ha, and 1.8 g Cd per ha in the belowground part. Punshon and Dickinson [67] suggested that if *Salix* spp. has enough genetic variability, the metal resistance can be regulated through acclimation. However, Beauchamp et al. [43] indicated that cuttings from contaminated areas did not benefit from this advantage when used in contaminated soil compared with cuttings collected from non-contaminated soils. Nevertheless, cuttings from contaminated areas should be recommended for future phytoremediation projects to increase their growth and survival.

## 5. Conclusions

In this study, all clones exhibited a large variation in heavy metal tolerance and accumulation capacity. In the current research, *S. jiangsuensis* '172' had a high survival rate and a relatively low capacity to concentrate heavy metals in their tissues in both tailings. Hence, this clone might have the potential to be used for phytostabilization programs

in heavily Pb/Zn- and Cu-polluted tailings in southern China. This study also found that the surviving cuttings of some clones could produce more biomass, such as *S. integra* 'WSH' (Pb/Zn tailing), *S. integra* 'HY', and 'WSH' (Cu tailing). However, these clones had a low survival rate in mine tailings with extremely high concentrations of heavy metals. Therefore, these clones could only fulfill the purpose of phytoremediation in moderate Pb/Zn or Cu tailings. In principle, all clones could easily take up and translocate Zn and Cd from the tailings to the aboveground parts, especially *S. integra*, while translocating Pb from the roots to the aboveground parts was limited. As stated above, these results could provide more complete information for phytoremediation potential and growth response using the stem cuttings of different clones cultivated directly in Pb/Zn and Cu tailings with extremely adverse environments. This study could also help in conducting simpler and low-cost phytoremediation methods for highly contaminated environments.

**Supplementary Materials:** The following supporting information can be downloaded at: https://www.mdpi.com/article/10.3390/f15020257/s1, Table S1. Comprehensive pollution index of heavy metals in mining tailings; Table S2. The recovery data for elemental analysis using inductively coupled plasma mass spectrometry; Table S3. Two-ANOVA of biomass, survival rate, CCI, and heavy metal concentration with treatment and clones; Table S4. Cutting biomass (g tree$^{-1}$) and ratio of belowground/aboveground of 14 *Salix* clones in different treatments after 15 months; Table S5. Chlorophyll concentration index (CCI) of leaves of 14 *Salix* clones in different treatments after 15 months; Table S6 Average heavy metal contents in aboveground parts of 14 *Salix* clones exposed to Pb/Zn tailings and Cu tailings (mg tree$^{-1}$); Table S7 Average heavy metal contents in belowground parts of 14 *Salix* clones exposed to Pb/Zn tailings and Cu tailings (mg tree$^{-1}$).

**Author Contributions:** C.J.: formal analysis, original draft preparation. Y.W. and C.M.: reviewing and editing. Y.C. conceptualization. S.W.: methodology. X.S.: conceptualization, methodology, funding acquisition. All authors have read and agreed to the published version of the manuscript.

**Funding:** This work was supported by the Key Scientific and Technological Grant of Zhejiang for Breeding New Agricultural Varieties (2016C02056-11).

**Data Availability Statement:** Data will be made available on request.

**Conflicts of Interest:** The authors declare that they have no conflicts of interest.

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
