# Peer review of "The Phytoremediation Potential of 14 Salix Clones Grown in Pb/Zn and Cu Mine Tailings"

_forests, doi:10.3390/f15020257_

Round 1

Reviewer 1 Report

Comments and Suggestions for Authors

This is an interesting manuscript. please see the following comments

1. quality of English should be urgently improved because now it is difficult to understand what is written

some very few examples are given here

replace

Fast-growing, huge biomass, deep roots, high transpiration rate, strong resilience to barren conditions, tolerance to excess 46 metals, and the ability to accumulate high levels of the metal in harvestable portions are 47 the characteristics of the optimal plant species for successful phytoremediation

with

Fast-growing plants that produce high amount of biomass, that are characterized by deep rooting system, high transpiration rate, strong resilience to barren conditions, tolerance to excess metals concentration, and that possess the ability to accumulate high levels of the metal in harvestable portions are the optimal plant species that can be used for successful phytoremediation

2. especially abstract which is the first impression that the reader has should be imprived. what is meant by h. The surviving cuttings of S. 18 integra ‘WSH’, S. matsudana ‘14’, S. chaenomeloides ‘3’, ‘4’, ‘5’ (Pb/Zn tailing); S. integra ‘HY’, ‘WSH’, 19 S. matsudana ‘14’, ‘19’, ‘34’ (Cu tailing)

what do the numbers correspond to? it makew no sense

you state

e up and translocate Zn and Cd from tailings to aboveground parts easily, 22 especially S. integra, while hardly translocate Pb and Cu from root to aboveground parts

why is Pb and Cu relevant here?

you state  71.40% and 85.70% of S. jiangsuensis ‘

what doe this % mean? out of 100 cuttings or something else? it is not clear

3. Introduction is quite long especially at the part where you talk about Salix. isnt it better to transfer this to the discussion? please erase and shorten considerably the introduction

4. you state land in China (GB 15618-2018) what does this  GB mean?

5.  You state The physicochemical properties of the mine tailings and river sand were analyzed (Table 1) It is not clear-did you do the analysis in this paper? then you should give the method of analysis or a reference that shows the method. If the analysis was done by somebody else you should aknowledge the people who did the analysis (eg the analysis was done in Institute A).

6. It is better to replace words with their corresponding symbols eg Pb, Zn, K throughout the manuscript

7. for all apparatuses eg pots, fabrics, etc and for all reagents manufacturer name, city and country of origin should be given

8. I suppose that for 2.6. Heavy Metal Determination in reference 45 information such as AAS model, limits of detection etc are given

9. In 2.5. Estimation of Chlorophyll Content about the e Opti-Sciences CCM-200 chlorophyll  content meter how reliable is this technique? is it based on some available protocol? please give more information 

10. In Fig 2 it is impossible to understand what each dot stands for. maybe a key at the beggining or at the end of the manuscript with the salix species and the corresponding code should be given. the same about the treatments what does T0, etc correspond to? the same about Fig 3

11.  I am not sure what is the use of Fig 4 since the graphs are so small. is there another way that will just show the most important accumulators? in any case the values are also found in Supplementary material

12. In the PCA graph what do the dots (green, red etc) correspond to?

13. the TOPSIS analysis you decsribe is very interesting but I did not understand is it correlated to the PCA analysis or is it completely separate from it?

14. please quote if possible Charvalas et al Determination of heavy metals in the territory of contaminated areas of Greece and their restoration through hyperaccumulators (2021) Environmental Science and Pollution Research, 28 (4), pp. 3858 - 3863, on the importance of bioremediation for mines

15. supplementary material should be after the reference list not before

Comments on the Quality of English Language

see main comments

Reviewer 2 Report

Comments and Suggestions for Authors

The review concerned an article entitled:  Phytoremediation Potential of Cutting of 14 Salix Clones Growing on Pb/Zn and Cu Mine Tailings.

The research focused on the potential of using Salix unrooted cuttings for phytoremediation of lead/zinc (Pb/Zn) and copper (Cu) mine tailings. Among 14 Salix clones tested, S. jiangsuensis ‘172’ exhibited high survival rates in Pb/Zn and Cu tailings. Other clones showed lower survival rates and varied biomass production. These findings offer valuable insights into willow growth and metal accumulation in challenging environments, supporting the use of cuttings for phytoremediation. I’m not fluent in English but I see also some language mistakes.

Questions and remarks:

Line 30. More keywords needed to localize article.

Introduction

Line 35.What are the potential hazards associated with the generation of trace elements during mining and smelting processes?

Line 40. Its also important to mention that plants have some resistance processes against heavy metals like antioxidant defence system described example in: 10.1038/s41598-021-82391-1.

Line 118 How were the plant materials selected for the experiment, and what characteristics were considered in choosing Salix clones for phytoremediation?

Line 126 Can you provide details on the experimental setup, including the greenhouse conditions, pot specifications, and the duration of the cultivation period for the Salix cuttings?

Line 146 What methods were employed for biomass measurements, chlorophyll content estimation, and heavy metal determination in both aboveground and belowground parts of the Salix clones? Additionally, how were bioconcentration factor (BCF) and translocation factor (TF) values calculated, and what significance do these values hold in the context of phytoremediation efficacy?

Line 425 Can you discuss the potential mechanisms or adaptive strategies employed by the surviving Salix jiangsuensis '172' in coping with the extremely high concentrations of heavy metals in mine tailings, and how do these mechanisms contribute to its high tolerance capacity for phytoremediation purposes?

Discussion:

1.     In the discussion on the limitations and challenges of phytoremediation in mine tailings, what additional factors, aside from pH and nutrient deficiencies, might influence the bioavailability of heavy metals in soil, and how do these factors impact the overall success of phytoremediation efforts?

2.     In the context of the potential for phytoremediation, could you elaborate on the specific recommendations for the utilization of Salix jiangsuensis '172' in mine tailings afforestation and ecological restoration, and how could the study findings inform future protocols for phytoremediation in highly contaminated areas?

3.     In the final section, the study recommends the utilization of thick cuttings (>1 cm diameter) in future phytoremediation protocols. What specific advantages do thicker cuttings offer in terms of survival rate, biomass production, and overall effectiveness in extremely high concentrations of heavy metals in tailings, and how might these recommendations be practically implemented in future phytoremediation initiatives?

Line 547 What are the key implications of the findings for future phytoremediation projects, and how might the identified characteristics of Salix clones guide the selection and implementation of suitable species or clones in diverse contaminated environments?

Round 2

Reviewer 1 Report

Comments and Suggestions for Authors

The manuscript has been improved

please see the following comments

see correct citing of Determination of heavy metals in the territory of contaminated areas of Greece and their restoration through hyperaccumulators

from https://doi.org/10.1007/s11356-020-11920-8

2) you have stated The acronym "GB" represents the national standard for pinyin. I am sorry I still cannot understand the meaning. the sentence was

 revealing that the  concentrations of heavy metals in the mine tailings exceeded the risk screening values for  soil contamination of agricultural land in China (GB 15618-2018) Is this some national standard limits according to legislation? if so it should be given in the list of references irrespective of language of publication

3) please proofread one last time trying to correct small inaccurracies in language

Comments on the Quality of English Language

please proofread one last time trying to correct small inaccurracies in language
